

# Structure and stability of recombinant bovine odorant-binding protein: I. Design and analysis of monomeric mutants

Olga V. Stepanenko[1], Denis O. Roginskii[1], Olesya V. Stepanenko[1], Irina M. Kuznetsova[1], Vladimir N. Uversky[1,3] and Konstantin K. Turoverov[1,2]

[1] Laboratory of Structural Dynamics, Stability and Folding of Proteins, Institute of Cytology, Russian Academy of Science, St. Petersburg, Russia
[2] Peter the Great St. Petersburg Polytechnic University, St. Petersburg, Russia
[3] Department of Molecular Medicine, Morsani College of Medicine, University of South Florida, Tampa, FL, United States

Corresponding authors
Vladimir N. Uversky,
vuversky@health.usf.edu
Konstantin K. Turoverov,
kkt@incras.ru

## ABSTRACT

Bovine odorant-binding protein (bOBP) differs from other lipocalins by lacking the conserved disulfide bond and for being able to form the domain-swapped dimers. To identify structural features responsible for the formation of the bOBP unique dimeric structure and to understand the role of the domain swapping on maintaining the native structure of the protein, structural properties of the recombinant wild type bOBP and its mutant that cannot dimerize via the domain swapping were analyzed. We also looked at the effect of the disulfide bond by designing a monomeric bOBPs with restored disulfide bond which is conserved in other lipocalins. Finally, to understand which features in the microenvironment of the bOBP tryptophan residues play a role in the defining peculiarities of the intrinsic fluorescence of this protein we designed and investigated single-tryptophan mutants of the monomeric bOBP. Our analysis revealed that the insertion of the glycine after the residue 121 of the bOBP prevents domain swapping and generates a stable monomeric protein bOBP-Gly121+. We also show that the restored disulfide bond in the GCC-bOBP mutant leads to the noticeable stabilization of the monomeric structure. Structural and functional analysis revealed that none of the amino acid substitutions introduced to the bOBP affected functional activity of the protein and that the ligand binding leads to the formation of a more compact and stable state of the recombinant bOBP and its mutant monomeric forms. Finally, analysis of the single-tryptophan mutants of the monomeric bOBP gave us a unique possibility to find peculiarities of the microenvironment of tryptophan residues which were not previously described.

## INTRODUCTION

Lipocalins constitute a family of carrier proteins that transport various small hydrophobic molecules ranging from lipids to retinoids, steroids, and bilins. Being found in animals, plants, and bacteria and possessing low sequence identity (less than 20%), these proteins are

characterized by the presence of the conserved "lipocalin fold" that includes two structural modules, an eight-stranded $\beta$-barrel that constitutes 70–80% of the protein and includes the ligand-binding site and a C-terminal $\alpha$-helix with unknown function (*Flower, North & Sansom, 2000*). Evolution of the lipocalin fold generated numerous specialized carrier proteins with the highly diversified binding specificities.

One of the sub-classes of the lipocalin family includes odorant binding proteins (OBPs) with the characteristic ability of reversible binding of various odorant molecules; i.e. volatile, small and hydrophobic compounds with no fixed structure and chemical properties (*Tegoni et al., 2000*). Classic odorant binding protein (OBP) is characterized by a specific monomeric fold, where the eight $\beta$-strands, a short $\alpha$-helical region, and the ninth $\beta$-strand interact to form a $\beta$-barrel followed by the disordered C-terminal tail (*Bianchet et al., 1996*; *Flower, North & Sansom, 2000*). The ligand binding site of these proteins is formed by hydrophobic and aromatic residues located within the inner cavity of the $\beta$-barrel and loop regions connecting $\beta$-strands of the barrel. The conserved disulfide bridge formed by Cys 63 and Cys 155 is commonly found in many OBPs to link the flexible C-terminal moiety and strand $\beta$4.

Curiously, despite rather high sequence identity between porcine and bovine OBPs (42%), these lipocalins are characterized by different quaternary structures, with porcine OBP (pOBP) being a monomeric protein (*Spinelli et al., 1998*), and with bovine OBP (bOBP) being a dimer (*Bianchet et al., 1996*; *Tegoni et al., 1996*), protomers of which lack the disulfide bridge which is a common feature for all lipocalin family members (*Tegoni et al., 2000*). Therefore, bOBP has a unique dimeric structure, which is quite different from the monomeric folds of the majority of classical OBPs (*Bianchet et al., 1996*; *Stepanenko et al., 2014b*) (Fig. 1). In the bOBP dimer, each of the two protomers forms a $\beta$-barrel that interacts with the $\alpha$-helical portion of the C-terminal tail of other protomer via the domain swapping mechanism. Such a mechanism was described for many dimeric and oligomeric protein complexes, where it plays important structural and functional roles (*Bennett, Schlunegger & Eisenberg, (1995)*; *Van der Wel, (2012)*). It is believed that the increased interaction area between the protein subunits in the complexes formed via the domain swapping mechanism affects the overall protein stability (*Bennett, Choe & Eisenberg, 1994*; *Liu & Eisenberg, 2002*). In some cases, the formation of the quaternary structure of protein by this mechanism is associated with the emergence of new functions in protein oligomers which are not found in the monomeric forms of these proteins (*Liu & Eisenberg, 2002*). Finally, the domain swapping mechanism is involved in the early stages of the amyloid fibril formation (*Van der Wel, 2012*).

In this work, to identify which structural features of bOBP are responsible for the formation of its unique dimeric structure and to understand the role of the domain swapping mechanism in maintaining the native structure of the protein, structural properties of the recombinant wild type bOBP and its four mutant forms that cannot dimerize via the domain swapping (*Ramoni et al., 2008*; *Ramoni et al., 2002*) were analyzed and compared using a spectrum of the biophysical techniques that included intrinsic fluorescence spectroscopy, circular dichroism spectroscopy in the far- and near-UV regions and gel filtration. We also designed two monomeric mutants, GCC-bOBP-W17F and
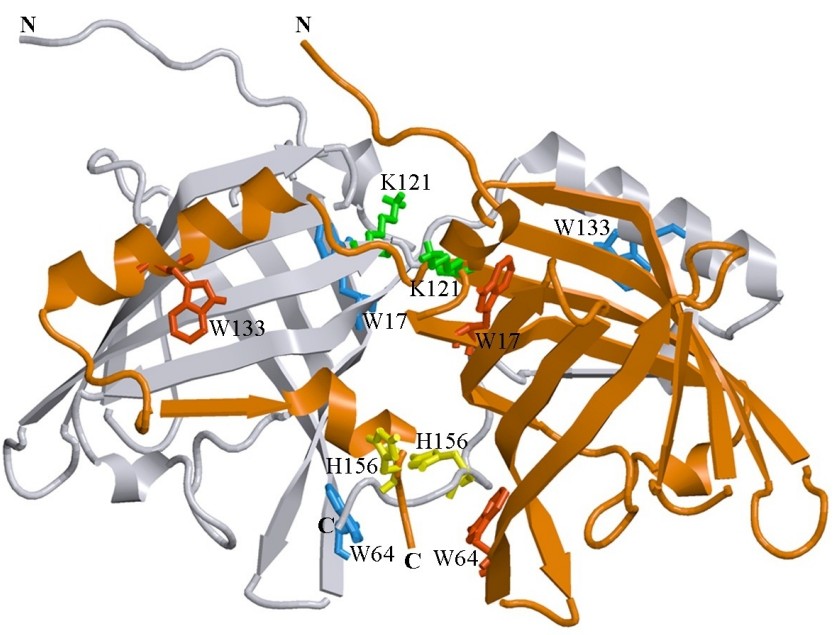

**Figure 1** **3D structure of bOBP.** The individual subunits in the protein are in gray and orange. The tryptophan residues in the different subunits are indicated in red and blue. The Lys 121 residue after which an extra glycine residue are inserted in the mutant form bOBP-Gly121+ is drown in green. Additionally the residues Trp 64 and His 156 (yellow) are substituted for cysteine in the mutant form GCC-bOBP. The drawing was generated based on the 1OBP file (*Tegoni et al., 1996*) from PDB (*Dutta et al., 2009*) using the graphic software VMD (*Hsin et al., 2008*) and Raster3D (*Merritt & Bacon, 1977*).

GCC-bOBP-W133F, each containing a single tryptophan residue, for the characterization of the specific features of the microenvironments of these tryptophan residues that affect the intrinsic fluorescence characteristics of this protein.

## MATERIALS AND METHODS

### Materials

GdnHCl (Nacalai Tesque, Japan) and 1-octen-3-ol (OCT; Sigma-Aldrich, St. Louis, MO, USA) were used without further purification. The protein concentration was 0.1–0.2 mg/ml. The OCT concentration was 10 mM. The experiments were performed in 20 mM Na-phosphate-buffered solution at pH 7.8.

### Gene expression and protein purification

The plasmids pT7-7-bOBP which encodes recombinant bOBP and its mutant forms with a poly-histidine tag were used to transform *Escherichia coli* BL21(DE3) host (Invitrogen) (*Stepanenko et al., 2014b*). The plasmids encoding bOBP and its mutant forms were purchased from Evrogen JSC. Protein expression was induced by incubating the cells with 0.3 mM of isopropyl-beta-D-1-thiogalactopyranoside (IPTG; Fluka, Buchs, Switzerland) for 24 h at 37 °C. The recombinant protein was purified with Ni+-agarose packed in HisGraviTrap columns (GE Healthcare, Uppsala, Sweden). The protein purity was determined through SDS-PAGE in 15% polyacrylamide gel (*Laemmli, 1970*).

## Analyzing the 3D protein structure

We analyzed the microenvironment peculiarities for tryptophan residues in the bOBP and GCC-bOBP structure based on PDB data (*Dutta et al., 2009*) using the 1OBP (*Tegoni et al., 1996*) and 2HLV PDB files (*Ramoni et al., 2008*) as described previously (*Giordano et al., 2004; Stepanenko et al., 2014a; Stepanenko et al., 2015; Stepanenko et al., 2012; Turoverov, Kuznetsova & Zaitsev, 1985*).

## Fluorescence spectroscopy

Fluorescence experiments were performed using a Cary Eclipse spectrofluorometer (Varian, Australia) with microcells FLR (10 × 10 mm; Varian, Australia). Fluorescence intensity was corrected for the primary inner filter effect (*Fonin et al., 2014*). Fluorescence lifetime were measured using a "home built" spectrofluorometer with nanosecond impulse (*Turoverov et al., 1998*) as well as micro-cells (101.016-QS 5 × 5 mm; Hellma, Germany). The decay curves were analyzed using earlier elaborated program (*Turoverov et al., 1998*).

The fitting routine was based on the non-linear least-squares method. Minimization was accomplished according to (*Marquardt, 1963*). P-terphenyl in ethanol and N-acetyl-tryptophanamide in water were used as reference compounds (*Zuker et al., 1985*). Experimental data were analyzed using the multiexponential approach:

$$I(t) = \sum_i \alpha_i \exp(-t/\tau_i)$$

where $\alpha_i$ and $\tau_i$ are amplitude and lifetime of component $i$, respectively, and were $\sum \alpha_i = 1$. At the same time, more physical sense (*Turoverov & Kuznetsova, 1986*) is given by the contribution of $i$ component (with $\tau_i$) $S_i$ to the total emission:

$$S_i = \frac{\alpha_i \int_0^\infty \exp(-t/\tau_i)\,dt}{\sum \alpha_i \int_0^\infty \exp(-t/\tau_i)\,dt} = \frac{\alpha_i \tau_i}{\sum \alpha_i \tau_i}.$$

The root-mean square value of fluorescent lifetimes, $\langle \tau \rangle$, for biexponential decay was determined as

$$\langle \tau \rangle = \frac{\alpha_1 \tau_1^2 + \alpha_2 \tau_2^2}{\alpha_1 \tau_1 + \alpha_2 \tau_2} = \sum S_i \tau_i.$$

Tryptophan fluorescence in the protein was excited at the long-wave absorption spectrum edge ($\lambda_{ex} = 297$ nm), wherein the tyrosine residue contribution to the bulk protein fluorescence is negligible. The fluorescence spectra position and form were characterized using the parameter $A = I_{320}/I_{365}$, wherein $I_{320}$ and $I_{365}$ are the fluorescence intensities at the emission wavelengths 320 and 365 nm, respectively (*Turoverov & Kuznetsova, 2003*). The values for parameter $A$ and the fluorescence spectrum were corrected for instrument sensitivity. The tryptophan fluorescence anisotropy was calculated using the equation:

$$r = (I_V^V - GI_H^V)/(I_V^V + 2GI_H^V),$$

wherein $I_V^V$ and $I_H^V$ are the vertical and horizontal fluorescence intensity components upon excitement by vertically polarized light. $G$ is the relationship between the fluorescence

intensity vertical and horizontal components upon excitement by horizontally polarized light ($G = I_V^H / I_H^H$), $\lambda_{em} = 365$ nm (*Turoverov et al., 1998*).

Protein unfolding was initiated by manually mixing the protein solution (40 μL) with a buffer solution (510 μL) that included the necessary GdnHCl concentration. The GdnHCl concentration was determined by the refraction coefficient using an Abbe refractometer (LOMO, Russia; (*Pace, 1986*)). The dependences of different bOBP fluorescent characteristics on GdnHCl were recorded following protein incubation in a solution with the appropriate denaturant concentration at 4 °C for 2, 24 and 48 h. bOBP refolding was initiated by diluting the pre-denatured protein (in 3.0 M GdnHCl, 40 μL) with the buffer or denaturant solutions at various concentrations (510 μL). The spectrofluorometer was equipped with a thermostat that holds the temperature constant at 23 °C.

## Circular dichroism measurements

The CD spectra were generated using a Jasco-810 spectropolarimeter (Jasco, Japan). Far-UV CD spectra were recorded in a 1-mm path length cell from 260 nm to 190 nm with a 0.1 nm step size. Near-UV CD spectra were recorded in a 10-mm path length cell from 320 nm to 250 nm with a 0.1 nm step size. For the spectra, we generated three scans on average. The CD spectra for the appropriate buffer solution were recorded and subtracted from the protein spectra.

## Gel filtration experiments

We performed gel filtration experiments for bOBP and its mutant forms in a buffered solution without and with addition of GdnHCl using a Superdex-75 PC 3.2/30 column (GE Healthcare, Sweden) and an AKTApurifier system (GE Healthcare, Uppsala, Sweden). The column was equilibrated with the buffered solution or GdnHCl at the desired concentration, and 10 μl of the protein solution prepared under the same conditions was loaded on the pre-equilibrated column. The change in hydrodynamic dimensions for the studied proteins was evaluated as a change in the bOBP or the mutant protein elution volume. Multiple proteins with known molecular masses (aprotinin (6.5 kDa), ribonuclease (13.7 kDa), carbonic anhydrase (29 kDa), ovalbumin (43 kDa) and conalbumin (75 kDa), which are chromatography standards from GE Healthcare) were used to generate the calibration curve.

## Evaluation of the intrinsic disorder predisposition

The intrinsic disorder propensity of the bOBP was evaluated by several disorder predictors, such as PONDR® VLXT (*Dunker et al., 2001*), PONDR® VSL2 (*Peng et al., 2005*), PONDR® VL3 (*Peng et al., 2006*), and PONDR® FIT (*Xue et al., 2010*). Effects of the point mutations on the intrinsic disorder predisposition of this protein was analyzed by PONDR® VSL2. In these analyses, scores above 0.5 are considered to correspond to the disordered residues/regions. PONDR® VSL2B was chosen for the comparative analysis of the bOBP mutants since this tool is one of the more accurate stand-alone disorder predictors (*Fan & Kurgan, 2014*; *Peng & Kurgan, 2012*), PONDR® VLXT is known to have high sensitivity to local sequence peculiarities and can be used for identifying disorder-based interaction sites (*Dunker et al., 2001*), PONDR® VL3 provides accurate evaluation of long disordered
regions (*Peng et al., 2006*), whereas a metapredictor PONDR-FIT is moderately more accurate than each of the component predictors, PONDR® VLXT (*Dunker et al., 2001*), PONDR® VSL2 (*Peng et al., 2005*), PONDR® VL3 (*Peng et al., 2006*), FoldIndex (*Prilusky et al., 2005*), IUPred (*Dosztanyi et al., 2005*), TopIDP (*Campen et al., 2008*). PONDR-FIT (*Xue et al., 2010*).

## RESULTS AND DISCUSSION

It is believed that the introduction of an extra glycine residue after the bOBP residue 121 (Fig. 2) should result in the increased mobility of the loop connecting $\alpha$-helix and 8th $\beta$-strand of the $\beta$-barrel, which, in its turn, promotes the formation of a monomeric fold of the mutant protein bOBP-Gly121+. Substitutions of the residues Trp64 and His156 to cysteines in bOBP-Gly121+ generate a mutant form GCC-bOBP, which should have stable monomeric structure due to the restoration of the disulfide bond typically seeing in classical OBPs. Finally, to characterize specific features of the microenvironments of tryptophan residues W17 and W133 that might affect the intrinsic fluorescence of the protein, we designed two monomeric mutant forms GCC-bOBP-W17F and GCC-bOBP-W133F, each containing a single tryptophan residue.

To evaluate potential effects of selected mutations on protein structure, we analyzed substitution-induced changes in the intrinsic disorder propensity of bOBP. It has been pointed out that such computational analysis can provide useful information on the expected outcomes of the point mutations in proteins (*Melnik et al., 2012*; *Moroz et al., 2013*; *Uversky et al., 2011*; *Vacic et al., 2012*). Figure 3A represents the results of the computational multi-tool analysis of the per-residue intrinsic disorder predisposition of bOBP. We used here several members of the PONDR family, PONDR® VLXT (*Dunker et al., 2001*), PONDR® VSL2 (*Peng et al., 2005*), PONDR® VL3 (*Peng et al., 2006*), and PONDR-FIT (*Xue et al., 2010*). Figure 3A shows that all these tools are generally agree with each other and indicates that although bOBP is predicted to be mostly ordered, this protein possesses several disordered or flexible regions. Disordered regions are defined here as protein fragments containing residues with the disorder scores above the 0.5 threshold, whereas regions are considered flexible if disorder scores of their residues ranges from 0.3 to 0.5. Figure 3B represents the results of the disorder evaluation in mutant forms of the bOBP and shows the aligned PONDR® VSL2-based disorder profiles for the wild type protein and its four mutants. These analyses revealed that the wild type bOBP and its four mutants are expected to be rather ordered (clearly belonging to the category of hybrid proteins that contain ordered domains and intrinsically disordered regions) and that mutations do not induce significant changes in the bOBP disorder propensity.

Previously, we have shown that the recombinant bOBP, unlike native bOBP purified from the tissue, exists in a stable native-like state as a mixture of monomeric and dimeric forms (*Stepanenko et al., 2014b*) (Table 1). Furthermore, the recombinant bOBP forms dimers in the presence of relatively high denaturant concentrations (e.g., in a solution of 1.5 M guanidine hydrochloride, GdnHCl). The dimerization process is accompanied by the formation of a stable, more compact, intermediate state maximally populated at 0.5M GdnHCl.

**1. bOBPwt** – wild type protein forms dimer via the domain-swapping mechanism. The protein has 3 tryptophan residues.

```
1          10         20         30         40         50         60
AQEEEAEQNL SELSGPWRTV YIGSTNPEKI QENGPFRTYF RELVFDDEKG TVDFYFSVKR
61         70         80         90         100        110        120
DGKWKNVHVK ATKQDDGTYV ADYEGQNVFK IVSLSRTHLV AHNINVDKHG QTTELTGLFV
121        130        140        150        159
KLNVEDEDLE KFWKLTEDKG IDKKNVVNFL ENEDHPHPE
```

**2. bOBP-Gly121+** – the introduction of an extra glycine residue after the bOBP residue 121 is proposed to inhibit dimer formation as a result of the increased mobility of the loop connecting α-helix and 8th β-strand of the β-barrel. The protein contains 3 tryptophan residues as well.

```
1          10         20         30         40         50         60
AQEEEAEQNL SELSGPWRTV YIGSTNPEKI QENGPFRTYF RELVFDDEKG TVDFYFSVKR
61         70         80         90         100        110        120
DGKWKNVHVK ATKQDDGTYV ADYEGQNVFK IVSLSRTHLV AHNINVDKHG QTTELTGLFV
121        130        140        150        159
KGLNVEDEDLE KFWKLTEDKG IDKKNVVNFL ENEDHPHPE
 121+
```

**3. GCC- bOBP (bOBP-Gly121+-W64C-H155C)** – the substitutions W64C and H155C result in the restoration of the disulfide bond which is necessary for the additional stabilization of the protein. The protein has only 2 tryptophan residues.

```
1          10         20         30         40         50         60
AQEEEAEQNL SELSGPWRTV YIGSTNPEKI QENGPFRTYF RELVFDDEKG TVDFYFSVKR
61         70         80         90         100        110        120
DGKCKNVHVK ATKQDDGTYV ADYEGQNVFK IVSLSRTHLV AHNINVDKHG QTTELTGLFV
121        130        140        150        159
KGLNVEDEDLE KFWKLTEDKG IDKKNVVNFL ENEDCPHPE
 121+
```

**4. GCC-bOBP-W17F (bOBP-Gly121+-W64C-H155C-W17F)** – the protein contains a single tryptophan residue which allows the investigation of the features of the microenvironment of this residue.

```
1          10         20         30         40         50         60
AQEEEAEQNL SELSGPFRTV YIGSTNPEKI QENGPFRTYF RELVFDDEKG TVDFYFSVKR
61         70         80         90         100        110        120
DGKCKNVHVK ATKQDDGTYV ADYEGQNVFK IVSLSRTHLV AHNINVDKHG QTTELTGLFV
121        130        140        150        159
KGLNVEDEDLE KFWKLTEDKG IDKKNVVNFL ENEDCPHPE
 121+
```

**5 GCC-bOBP-W133F (bOBP/Gly121+/W64C/H155C/W133F)** – the protein has a single tryptophan residue as well.

```
1          10         20         30         40         50         60
AQEEEAEQNL SELSGPWRTV YIGSTNPEKI QENGPFRTYF RELVFDDEKG TVDFYFSVKR
61         70         80         90         100        110        120
DGKCKNVHVK ATKQDDGTYV ADYEGQNVFK IVSLSRTHLV AHNINVDKHG QTTELTGLFV
121        130        140        150        159
KGLNVEDEDLE KFFKLTEDKG IDKKNVVNFL ENEDCPHPE
 121+
```

**Figure 2** **Sequence peculiarities of various bPDB forms.** The comparison of the primary sequence for the recombinant bOBPwt and its mutant forms bOBP-Gly121+ and GCC- bOBP, which are not able to form domain-swapped dimers. The mutant forms GCC-bOBP-W17F and GCC-bOBP-W133F designed to contain a single tryptophan residue were produced to investigate the peculiarities of the microenvironment of the tryptophan residues.

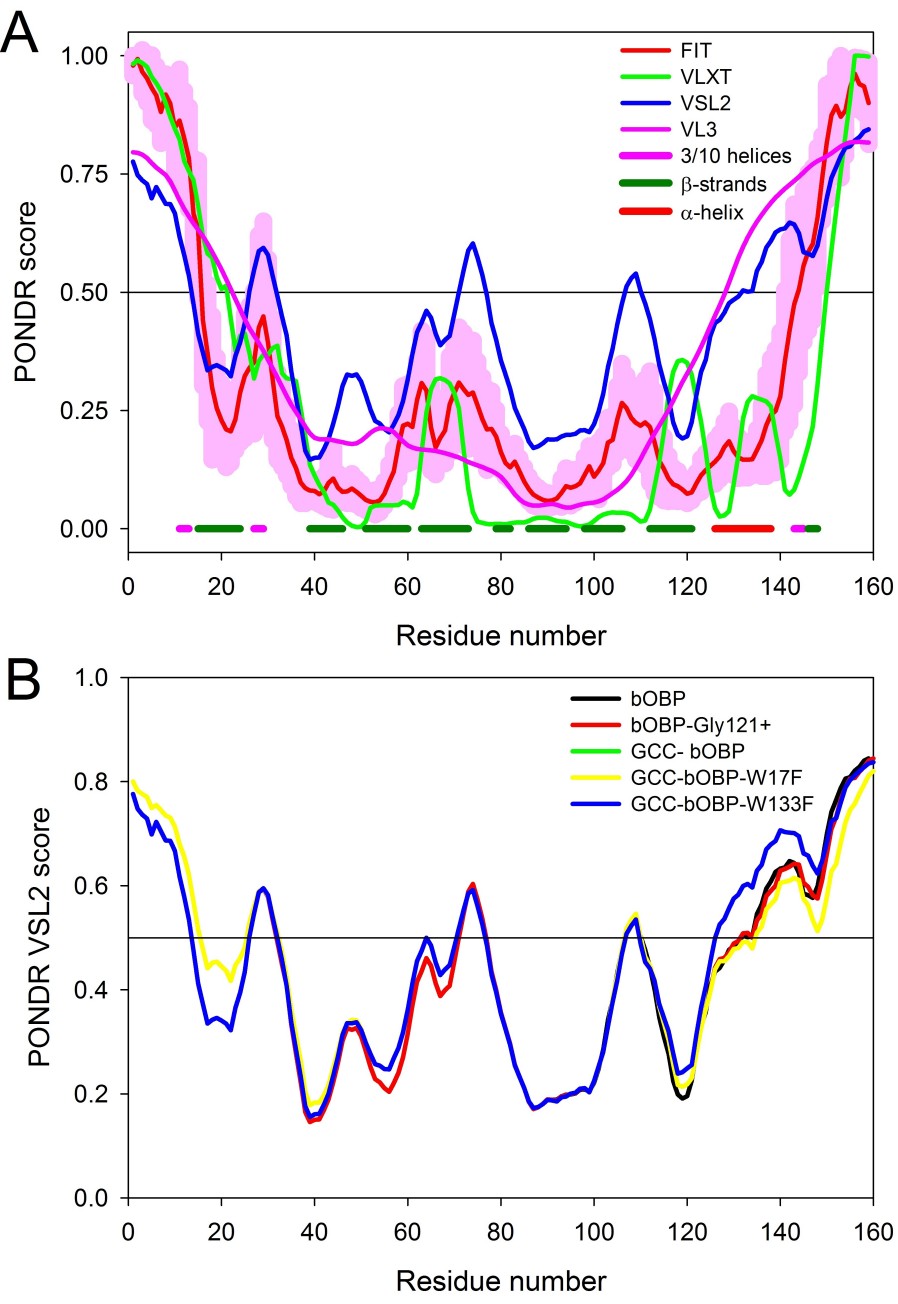

**Figure 3   Intrinsic disorder propensity of the wild type bOBP and its mutants.** (A) Per-residue disorder propensity of the wild type bOBP evaluated by members of the PONDR family, PONDR® VLXT (*Dunker et al., 2001*) (green line), PONDR® VSL2 (*Peng et al., 2005*) (blue line), PONDR® FIT (*Xue et al., 2010*) (red line) and PONDR® VL3 (*Peng et al., 2006*) (pink line). Localization of known elements of the bOBP secondary structure is shown by colored bars at the bottom of the plot. Light pink shadow around the PONDR® FIT curve represents distribution of errors in the disorder score evaluation. (B) Effects of mutations on the intrinsic disorder propensity of bOBP evaluated by PONDR® VSL2.
**Table 1** Characteristics of recombinant bOBPwt and its mutant forms in different structural states as well as in the presence of natural ligand OCT.

| | Intrinsic fluorescence | | | | Hydrodynamic dimensions | |
|---|---|---|---|---|---|---|
| | $\lambda_{max}$, nm[a] | Parameter $A$[a] | $r$[b] | $\langle\tau\rangle$, ns:[c] $S_i$; $\tau_i$ (ns) | 1 peak, kDa | 2 peak, kDa |
| bOBPwt (*in buffered solution*)[d] | 335 | 1.21 | 0.170 | 4.4 ± 0.2: 0.45; 2.7 0.55; 6.0 | 43.9 | 23.8 |
| bOBPwt (*in 0.5 M GdnHCl-I$_1$ state*)[d] | 337 | 1.10 | 0.166 | 4.6 ± 0.2: 0.30; 2.6 0.70; 5.3 | 34.0 | 19.3 |
| bOBPwt (*in 1.6 M GdnHCl-I$_2$ state*)[d] | 335 | 1.20 | 0.180 | 4.8 ± 0.1: 0.43; 2.5 0.57; 6.4 | 43.6 | |
| bOBPwt/OCT | 334 | 1.30 | 0.177 | | 39.6 | 21.5 |
| bOBPwt in 3.5 M GdnHCl | 349 | 0.47 | 0.062 | | | |
| bOBP/Gly121+ | 336 | 1.13 | 0.166 | 4.6 ± 0.1: 0.14; 1.6 0.86; 5.1 | 23.6 | |
| bOBP/Gly121+/OCT | 335 | 1.12 | 0.170 | | 21.5 | |
| bOBP/Gly121+ in 3.5 M GdnHCl | 350 | 0.47 | 0.059 | | | |
| GCC-bOBP | 335 | 1.05 | 0.170 | 4.3 ± 0.2: 0.45; 2.9 0.55; 5.4 | 23.6 | |
| GCC-bOBP/OCT | 335 | 1.05 | 0.174 | | 22.5 | |
| GCC-bOBP in 3.5 M GdnHCl | 348 | 0.48 | 0.060 | | | |
| GCC-bOBP-W17F | 339 | 0.82 | 0.164 | 4.7 ± 0.1: 0.75; 3.8 0.25; 7.1 | 23.6 | |
| GCC-bOBP-W17F/OCT | 339 | 0.82 | 0.165 | | 22.5 | |
| GCC-bOBP-W17F in 3.5 M GdnHCl | 349 | 0.49 | 0.066 | | | |
| GCC-bOBP-W133F | 325 | 2.83 | 0.186 | 1.9 ± 0.4: 0.73; 0.20 0.27; 4.3 | 23.6 | |
| GCC-bOBP-W133F/OCT | 323 | 3.02 | 0.189 | | 23.6 | |
| GCC-bOBP-W133F in 3.5 M GdnHCl | 350 | 0.46 | 0.056 | | | |

**Notes.**
[a] $\lambda_{ex} = 297$ nm.
[b] $\lambda_{ex} = 297$ nm, $\lambda_{em} = 365$ nm.
[c] $\lambda_{ex} = 297$ nm, $\lambda_{em} = 335$ nm.
[d] The data are from *Stepanenko et al. (2014b)*.

In the present work, gel filtration analysis revealed that all investigated mutants of the bOBP, namely bOBP-Gly121+, GCC-bOBP, GCC-bOBP-W17F, and GCC-bOBP-W133F, are monomers (Fig. 4, Table 1). The positions of the elution peaks of the studied mutant bOBP forms coincided with the position of the elution peak of the monomeric form of recombinant bOBP. This allows us to conclude that monomeric forms of mutant proteins bOBP-Gly121+, GCC-bOBP, GCC-bOBP-W17F, and GCC-bOBP-W133F are as compact

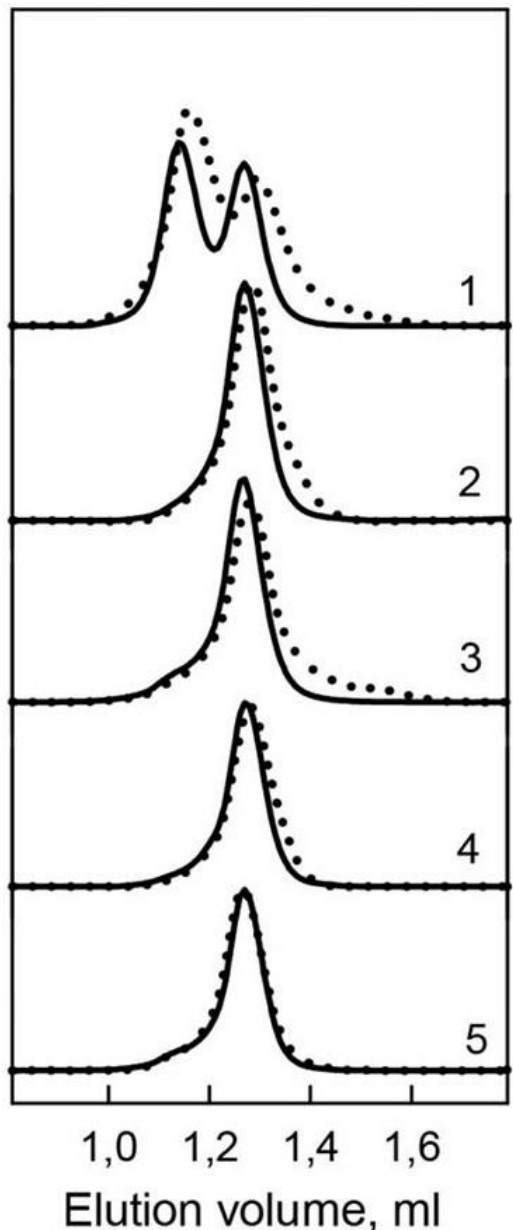

1,0   1,2   1,4   1,6
Elution volume, ml

**Figure 4  Hydrodynamic characteristics of the bOBP and its mutants.** The changes of hydrodynamic dimensions of recombinant bOBP (1) and its mutant forms bOBP-Gly121+ (2), GCC-bOBP (3), GCC-bOBP-W17F (4) and GCC-bOBP-W133F (5) in the absence (solid lines) and the presence of OCT (dotted lines).

as the recombinant bOBP monomer. Therefore, the amino acid substitutions introduced to the bOBP sequence did not affect the compact structure of this protein.

Investigation of the interaction of recombinant bOBP with its natural ligand 1-Octen-3-ol (OCT) by gel-filtration chromatography revealed that the elution profile of the bOBP/OCT complex contained two peaks (Fig. 4). These data indicate that similar to the recombinant bOBP the bOBP/OCT complex exists as a mixture of monomeric and dimeric

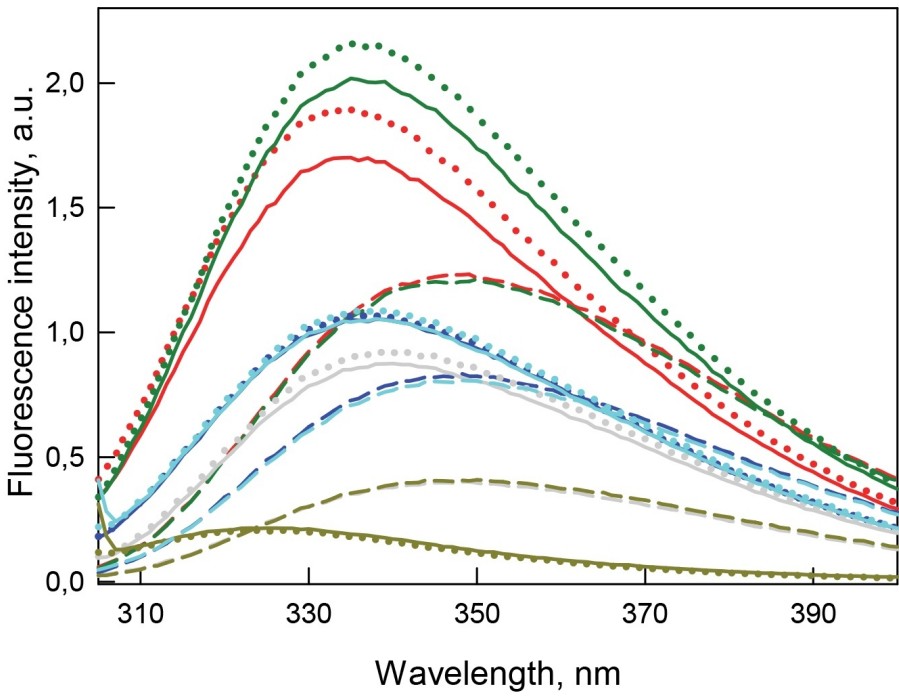

**Figure 5** **Tertiary structure changes for bOBP (red) and its mutant forms bOBP-Gly121+ (green), GCC-bOBP (blue), GCC-bOBP-W17F (gray) and GCC-bOBP-W133F (dark yellow) in different structural states are indicated by intrinsic tryptophan fluorescence ($\lambda_{ex} = 297$ nm).** The spectra shown are for the protein in buffered solution (solid line), in the presence of natural ligand OCT (dotted line) and in the presence of 3.5 M GdnHCl (dashed line). The corresponding spectra in light blue were obtained as a sum of the spectra for GCC-bOBP-W17F and GCC-bOBP-W133F.

forms of the protein. However, the positions of the two peaks in the bOBP/OCT elution profile are shifted to slightly higher elution volume, suggesting that the OCT binding induces partial compaction of both the monomeric and dimeric forms of the protein. Evaluated hydrodynamic dimensions of the bOBP and its complex with ligand confirmed this assumption (Table 1). Furthermore, the complexed forms of the bOBP mutants were also more compact than their OCT-free forms (Table 1). The similar behavior of recombinant bOBP and its mutants suggested that the introduced mutations do not affect functional activity of the bOBP and the ability of this protein to bind a natural ligand.

Table 1 and Fig. 5 shows that the recombinant bOBP is characterized by a relatively long-wave position of the intrinsic tryptophan fluorescence ($\lambda_{max} = 335$ nm at $\lambda_{ex} = 297$ nm). Although in comparison of the position of the emission spectrum of completely solvent accessible Trp ($\lambda_{max} = 350$ nm), the indicated value of 335 nm found for the recombinant bOBP is noticeably blue-shifted, this value is definitely too far from a short wavelength position of tryptophan fluorescence found in some proteins. In fact, one of the shortest wavelength position of Trp fluorescence ($\lambda_{max} = 308$ nm) was described for the blue copper protein azurine from *Pseudomonas aeruginosa* (*Turoverov, Kuznetsova & Zaitsev, 1985*), and a close short wavelength fluorescence spectra with $\lambda_{max} = 312$ nm and 318 nm were also described for the DsbC from *E.coli* (*Stepanenko et al., 2004*) and for another

**Table 2  Side chain conformation of Trp residues in bOBPwt and GCC-bOBP.**

| Protein | Residue | N ($d$)[a] | $\chi_1$, (deg)[a] | $\chi_2$, (deg)[a] |
|---------|---------|-----------|-----------|-----------|
| bOBPwt | Trp 17 | 84 (0.80) | 283.18 | 78.87 |
| | Trp 64 | 80 (0.71) | 278.04 | 99.85 |
| | Trp 133 | 56 (0.54) | 287.95 | 112.22 |
| GCC-bOBP | Trp 17 | 87 (0.83) | 292.40 | 82.09 |
| | Trp 133 | 48 (0.50) | 283.53 | 103.91 |

**Notes.**

[a]N is the number of atoms in the microenvironment of tryptophan residue; $d$ is the density of tryptophan residue microenvironment; $\chi_1$ and $\chi_2$ are the angles characterizing the conformation of tryptophan residue side chain.

cooper-containing bacterial protein amicyanin from *Thiobacillus versutus* (*Rosato et al., 1991*), respectively.

The intrinsic fluorescence of bOBP is determined by three tryptophan residues, two of which are located in the $\beta$-sheet (Trp17 is in the first $\beta$-strand, and Trp64 is in the fourth $\beta$-strand), whereas Trp133 is included into a single $\alpha$-helix of this protein. Among all the tryptophan residues of the protein Trp133 has the lowest density of the microenvironment (d = 0.54), indicating that it is partially accessible to the solvent (Table 2). The microenvironments of Trp17 and Trp64 are more dense (0.80 and 0.71, respectively), but also more polar compared with the Trp133 local environment (Tables 2–5).

It should be noted that the side chains of the charged residues Lys121 and Lys59 included in the microenvironments of Trp17 and Trp64, respectively, are oriented parallel to the indole ring of the corresponding tryptophan residue, and their NZ amino groups are located at a short distance from NE1 group of the corresponding tryptophan residue (5.16 and 4.55 Å for NZ groups Lys121 and Lys59, Tables 3 and 4). Therefore, the presence of a partial fluorescence quenching of Trp17 and Trp64 cannot be excluded, since fluorescence quenching was previously reported for a single tryptophan residue Trp16 of porcine OBP that has similar features in its microenvironment (*Staiano et al., 2007*; *Stepanenko et al., 2008*).

Recombinant bOBP is characterized by high values of fluorescence anisotropy and fluorescence lifetime (Table 1), and also has a pronounced CD spectrum in the near-UV region (Fig. 6). These observations indicate that the environment of tryptophan residues of this protein is quite rigid and asymmetric.

The monomeric bOBP-Gly121+ is characterized by a somewhat longer wavelength of the tryptophan fluorescence spectrum and lower value of the fluorescence anisotropy compared to the recombinant bOBP (Table 1, Fig. 5). The near-UV CD spectrum of the bOBP-Gly121+ is almost indistinguishable for the spectrum of recombinant bOBP (Fig. 6). This indicates that although the overall spatial structure of the protein is not perturbed by adding an extra Gly residue after the position 121, the local microenvironment of the tryptophan residues become less dense due to this sequence perturbation. Importantly, the magnitudes of the fluorescence lifetime and fluorescence quantum yield of the bOBP-Gly121+ are higher than those for the recombinant bOBP. It is likely that the more mobile microenvironments of the tryptophan residues in bOBP-Gly121+ might result in the

**Table 3  Characteristics of the Trp 17 microenvironment in bOBPwt and GCC-bOBP.**

| bOBPwt | | | GCC-bOBP | | |
|---|---|---|---|---|---|
| Atoms of the microenvironment | Atoms of TRP | R[a], Å | Atoms of microenvironment | Atoms of TRP | R[a], Å |
| *Atoms of polar groups* | | | | | |
| OG Ser 14 | NE1 | 6.86 | Ser 14 | NE1 | 6.61 |
| NE Arg 18 | O | 4.16 | Arg 18 | C | 6.17 |
| NH1 Arg 18 | O | 2.95 | Arg 18 | C | 6.46 |
| NH2 Arg 18 | O | 5.18 | OG1 Thr 19 | O | 6.16 |
| OE1 Glu 42 | N | 5.44 | NH1 Arg 41 | O | 6.26 |
| OE2 Glu 42 | N | 6.15 | OG Ser 95 | CH2 | 6.02 |
| OG Ser 95 | CH2 | 6.67 | OG1 Thr 97 | CH2 | 6.78 |
| ND1 His 98 | CZ3 | 6.87 | ND1 His 98 | CZ3 | 6.75 |
| NZ Lys 121 | NE1 | 5.16 | NZ Lys 121 | NE1 | 4.83 |
| O HOH 318A | CZ3 | 6.87 | O HOH 1006 | O | 4.90 |
| | | | HOH 1047 | C | 4.33 |
| | | | HOH 1066 | N | 5.20 |
| | | | HOH 1080 | CD1 | 6.62 |
| | | | HOH 1107 | O | 6.46 |
| *Atoms of peptide bonds* | | | | | |
| O Leu 13 | NE1 | 2.87 | O Leu 13 | NE1 | 2.74 |
| O Ser 14 | NE1 | 5.05 | O Glu 12 | NE1 | 6.62 |
| N Ser 14 | NE1 | 4.86 | O Ser 14 | NE1 | 5.11 |
| O Gly 15 | N | 3.66 | N Ser 14 | NE1 | 4.74 |
| N Gly 15 | NE1 | 4.43 | O Gly 15 | N | 3.08 |
| N Pro 16 | N | 3.54 | N Gly 15 | CD1 | 4.19 |
| O Pro 16 | N | 2.25 | N Pro 16 | N | 3.15 |
| N Arg 18 | C | 1.32 | O Pro 16 | N | 2.21 |
| O Arg 18 | C | 3.95 | N Arg 18 | C | 1.33 |
| O Phe 40 | O | 3.26 | O Arg 18 | C | 3.93 |
| N Phe 40 | O | 5.10 | N Thr 19 | C | 4.49 |
| N Glu 42 | O | 4.22 | O Thr 19 | C | 6.91 |
| O Glu 42 | CD1 | 5.73 | O Tyr 39 | O | 6.37 |
| N Leu 43 | CD1 | 4.02 | O Phe 40 | O | 3.24 |
| O Leu 43 | CD1 | 4.49 | N Phe 40 | O | 5.11 |
| O Ser 95 | CH2 | 4.14 | N Arg 41 | O | 3.64 |
| N Ser 95 | CH2 | 5.77 | O Arg 41 | O | 3.53 |
| N Arg 96 | CH2 | 4.71 | N Glu 42 | O | 4.07 |
| O Arg 96 | CH2 | 6.11 | O Glu 42 | CB | 5.54 |
| N Thr 97 | CH2 | 5.09 | N Leu 43 | CD1 | 3.86 |
| O Thr 97 | CZ3 | 3.76 | O Leu 43 | CD1 | 4.44 |
| O His 98 | CH2 | 4.10 | N Val 44 | CD1 | 6.03 |
| N His 98 | CZ3 | 4.14 | O Tyr 55 | O | 5.97 |
| N Leu 99 | CZ3 | 3.74 | O Ser 95 | CH2 | 3.55 |

**Table 3** (*continued*)

| bOBPwt | | | GCC-bOBP | | |
|---|---|---|---|---|---|
| **Atoms of the microenvironment** | **Atoms of TRP** | **Rᵃ, Å** | **Atoms of microenvironment** | **Atoms of TRP** | **Rᵃ, Å** |
| O Leu 99 | CZ3 | 5.57 | N Ser 95 | CH2 | 5.32 |
| O Phe 119 | CZ3 | 3.85 | N Arg 96 | CH2 | 5.48 |
| N Phe 119 | CE3 | 5.75 | O Arg 96 | CZ2 | 4.43 |
| O Val 120 | CA | 3.38 | N Thr 97 | CH2 | 4.59 |
| N Val 120 | CE3 | 3.92 | O Thr 97 | CZ3 | 4.10 |
| O Lys 121 | CA | 6.07 | O His 98 | CH2 | 3.67 |
| N Lys 121 | CE3 | 3.51 | N His 98 | CH2 | 3.82 |
| N Leu 122 | CA | 4.61 | N Leu 99 | CZ3 | 3.64 |
| O Leu 122 | O | 5.97 | O Leu 99 | CZ3 | 5.59 |
| | | | N Val 100 | CZ3 | 6.72 |
| | | | O Phe 119 | CZ3 | 3.53 |
| | | | N Phe 119 | CE3 | 5.67 |
| | | | O Val 120 | CA | 3.51 |
| | | | N Val 120 | CE3 | 3.73 |
| | | | O Lys 121 | CD2 | 6.58 |
| | | | N Lys 121 | CE3 | 3.77 |
| | | | N Gly 121+ | CZ3 | 6.22 |
| *Atoms of nonpolar groups and aromatic residues* | | | | | |
| CB Pro 16 | N | 3.49 | CB Pro 16 | N | 3.73 |
| CB Phe 40 | O | 3.74 | CB Phe 40 | O | 3.72 |
| CE1 Phe 45 | NE1 | 4.61 | CE1 Phe 45 | NE1 | 4.06 |
| CB His 98 | CZ3 | 5.32 | CB His 98 | CZ3 | 5.18 |
| CB Phe 119 | CE3 | 4.13 | CB Phe 119 | CE3 | 4.08 |
| CD2 Leu 13 | CZ2 | 4.17 | CB Leu 13 | CZ2 | 4.65 |
| CB Leu 43 | CD1 | 3.65 | CB Leu 43 | CD1 | 3.52 |
| CD1 Leu 94 | CH2 | 4.26 | CD1 Leu 94 | CH2 | 4.37 |
| CB Leu 99 | CZ3 | 3.80 | CB Leu 99 | CZ3 | 3.75 |
| CB Val 120 | CE3 | 5.51 | CB Val 120 | CZ3 | 5.26 |
| CB Lys 121 | CD2/CE2 | 3.83 | CD Lys 121 | NE1 | 3.61 |
| CD1 Leu 122 | C | 4.69 | | | |

**Notes.**

ᵃR is the minimal distance between a atom of residue of the microenvironment of tryptophan residue and the nearest atom of its indole ring.

removal of some potential quenching groups from the indole ring of these tryptophan residues, thereby leading to a weakening of the quenching effects.

The values of the fluorescence parameters such as the position of the maximal tryptophan fluorescence, fluorescence anisotropy, and fluorescence lifetime for the triple mutant GCC-bOBP, which was designed to have disulfide bond via substituting residues Trp64 and His156 of the bOBP-Gly121+ to the cysteine residues, were similar to these parameters recorded for the recombinant bOBP (Table 1, Fig. 5). The intensity of the negative band in the near-UV CD spectrum of the GCC-bOBP variant was greater than that of the

**Table 4** Characteristics of the Trp 64 microenvironment in bOBPwt.

| Atom of the microenvironment | Atom of TRP | R[a], Å |
|---|---|---|
| *Atoms of polar groups* | | |
| OH Tyr 39 | CH2 | 5.35 |
| OG Ser 57 | CE3 | 5.05 |
| NZ Lys 59 | NE1 | 4.55 |
| NZ Lys 63 | N | 6.58 |
| NZ Lys 65 | O | 5.23 |
| ND1 His 157 | CD1 | 3.27 |
| NE2 His 157 | CD1 | 4.22 |
| OE1 Glu 159 | CH2 | 5.16 |
| OE2 Glu 159 | CH2 | 4.56 |
| HOH 205A | CH2 | 4.04 |
| HOH 231A | CZ3 | 5.70 |
| HOH 247A | CH2 | 5.80 |
| HOH 282A | CD2 | 3.74 |
| HOH 289A | CB | 4.18 |
| HOH 298A | CZ3 | 3.56 |
| HOH 328A | NE1 | 2.88 |
| *Atoms of peptide bonds* | | |
| N Tyr 39 | CZ3 | 6.04 |
| O Tyr 39 | CZ3 | 6.78 |
| N Ser 57 | CZ3 | 5.96 |
| O Ser 57 | CZ3 | 4.20 |
| N Val 58 | CE3/CZ3 | 4.00 |
| O Val 58 | C | 3.67 |
| N Lys 59 | CE3/CZ3 | 3.78 |
| O Lys 59 | CA | 6.52 |
| N Arg 60 | N | 5.11 |
| O Arg 60 | N | 5.63 |
| N Lys 63 | N | 3.61 |
| O Lys 63 | N | 2.25 |
| N Lys 65 | C | 1.32 |
| O Lys 65 | C | 4.07 |
| O Pro 156 | CE2 | 6.63 |
| N His 157 | NE1 | 5.98 |
| O His 157 | CZ2 | 5.90 |
| N Pro 158 | CZ2 | 4.47 |
| O Pro 158 | CZ2 | 6.07 |
| N Glu 159 | CZ2 | 4.12 |
| O Glu 159 | NE1 | 3.31 |
| *Atoms of nonpolar groups and aromatic residues* | | |
| **CE1** Tyr 39 | CH2 | 3.88 |

**Table 4** (*continued*)

| Atom of the microenvironment | Atom of TRP | $R^a$, Å |
|---|---|---|
| CE1 His 155 | CE3 | 6.40 |
| **CE1** His 157 | CD2 | 3.52 |
| **CD** Pro 158 | CZ2 | 3.69 |
| **CB** Val 58 | CZ3 | 5.50 |
| **CB** Lys 63 | N | 3.23 |
| **CB** Lys 65 | C | 3.46 |

**Notes.**

[a]R is the minimal distance between a residue involved in the microenvironment of tryptophan residue and its indole ring.

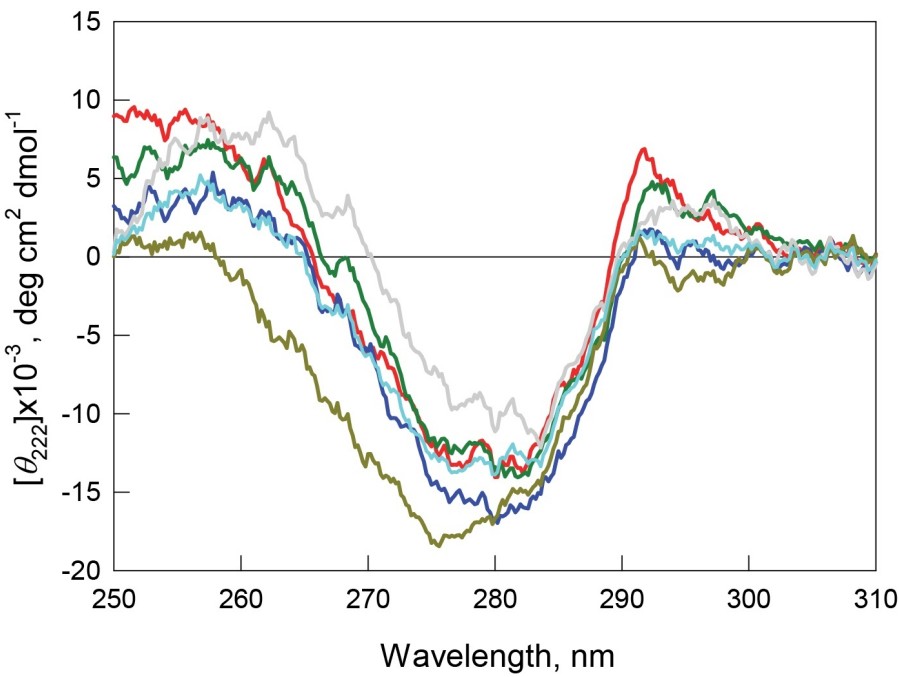

**Figure 6** **Tertiary structure for bOBP (red) and its mutant forms bOBP-Gly121+ (green), GCC-bOBP (blue), GCC-bOBP-W17F (gray) and GCC-bOBP-W133F (dark yellow) in buffered solution are indicated by near-UV CD spectra.** The spectrum in light blue was obtained as a sum of the spectra for GCC-bOBP-W17F and GCC-bOBP-W133F.

recombinant bOBP (Fig. 6). These data demonstrate the stabilizing effect of the disulfide bond to the protein structure. The intensity of the GCC-bOBP tryptophan fluorescence was approximately 25% lower than that of the recombinant bOBP (Fig. 5). Since the structure of the GCC-bOBP retained only two of the three tryptophan residues of the protein, namely Trp17 and Trp133, it can be argued that the removed residue Trp64 made a significant contribution to the fluorescence of recombinant bOBP.

Mutant forms designed to have a single tryptophan residue, GCC-bOBP-W17F (contains only Trp133) and the GCC-bOBP-W133F (contains only Trp17) are characterized by the substantially different parameters of tryptophan fluorescence and near-UV CD spectra (Figs. 5 and 6). GCC-bOBP-W17F has the most long-wavelength fluorescence spectrum

Stepanenko et al. (2016), *PeerJ*, DOI 10.7717/peerj.1933

**Table 5  Characteristics of the Trp 133 microenvironment in bOBPwt and GCC-bOBP.**

| bOBPwt | | | GCC-bOBP | | |
|---|---|---|---|---|---|
| Atoms of the microenvironment | Atom of TRP | $R^a$, Å | Atoms of the microenvironment | Atom of TRP | $R^a$, Å |
| *Atoms of polar groups* | | | | | |
| OH Tyr 21B | CZ2 | 4.36 | OH Tyr 21 | NE1 | 5.03 |
| OG1 Thr 136 | CA | 4.35 | OG1 Thr 136 | CA | 4.44 |
| NZ Lys 143 | CE3 | 3.35 | HOH 1087 | CH2 | 4.99 |
| HOH 218A | CE3/CZ3 | 4.69 | | | |
| HOH 2 54A | NE1 | 3.94 | | | |
| HOH 232B | NE1 | 3.27 | | | |
| HOH 283B | CZ2 | 5.83 | | | |
| *Atoms of peptide bonds* | | | | | |
| N Leu 129 | N | 6.23 | O Leu 129 | N | 2.97 |
| O Leu 129 | N | 3.10 | O Phe 132 | N | 2.26 |
| N Glu 130 | CD1 | 4.75 | N Lys 134 | C | 1.33 |
| O Glu 130 | N | 3.33 | | | |
| N Phe 132 | N | 2.85 | | | |
| O Phe 132 | N | 2.26 | | | |
| N Lys 134 | C | 1.33 | | | |
| O Lys 134 | O | 3.28 | | | |
| N Thr 136 | O | 3.37 | | | |
| O Thr 136 | O | 5.01 | | | |
| N Lys 143 | CZ3 | 5.56 | | | |
| O Lys 143 | CH2 | 5.36 | | | |
| N Val 146 | CH2 | 6.50 | | | |
| O Val 146 | CH2 | 6.63 | | | |
| *Atoms of nonpolar groups and aromatic residues* | | | | | |
| CE2 Tyr 21B | CZ2 | 4.17 | CE2 Tyr 21 | NE1 | 3.96 |
| C Phe 132 | N | 1.34 | C Phe 132 | N | 1.33 |
| C Leu 129 | CD1 | 3.90 | C Leu 129 | CD1 | 4.15 |
| CA Lys 134 | C | 2.43 | CG2Val 146 | CZ3 | 3.98 |
| CG2 Val 146 | CZ2 | 3.69 | CA Lys 143 | CZ3 | 4.55 |
| CG Lys 143 | CZ3 | 3.75 | | | |

**Notes.**

[a] R is the minimal distance between a residue involved in the microenvironment of tryptophan residue and its indole ring.

among all proteins studied here, the lowest values of the parameter $A$ and the fluorescence anisotropy $r$, whereas GCC-bOBP-W133F has the most short-wavelength fluorescence spectrum and the highest values of the parameter $A$ and fluorescence anisotropy $r$ (Table 1, Fig. 5). GCC-bOBP-W133F is also characterized by the most intense and short wavelength near-UV CD spectrum. At the same time, the near UV-CD spectrum of the GCC-bOBP-W17F is less intense and the most long wavelength among all proteins analyzed in this study (Fig. 6). These data indicate that the microenvironments of the residues Trp17 and Trp133 are significantly different from each other. It should be noted that the calculated
total spectrum of the intrinsic fluorescence of these two proteins GCC-bOBP-W17F and GCC-bOBP-W133F (calculated as a weighted sum of individual spectra) coincides with the tryptophan fluorescence spectrum of GCC-bOBP (Fig. 5). These data together with the results of the gel filtration analysis suggested that the mutant proteins GCC-bOBP-W17F and GCC-bOBP-W133F maintained native-like, mostly unperturbed spatial structures, and that the microenvironments of their residues Trp17 and Trp133 are similar to the environments of these residues in the GCC-bOBP protein.

The position of the tryptophan fluorescence spectrum of the GCC-bOBP-W17F mutant, and the values of its parameter $A$, fluorescence anisotropy, and fluorescence lifetime suggest that the microenvironment of the Trp133 residue is relatively polar and mobile, and the residue itself contributes significantly to the total fluorescence of this protein (Table 1). These data agree well with the results of the analysis of the specific characteristics of the microenvironment of tryptophan residues in the wild type bOBP (Tables 2–5) and the GCC-bOBP mutant (Tables 2–5). At the same time, the spectral characteristics of the GCC-bOBP-W133F mutant suggest that the Trp17 can be considered as an internal residue located within a very dense, inaccessible to solvent microenvironment. Furthermore, the fluorescence of this residue is substantially quenched (Table 1, Fig. 5).

Analysis of the microenvironment of tryptophan residue Trp17 in the wild type bOBP and its GCC-bOBP mutant (Table 3) shows that it is rather similar to the environment of Trp16 in pOBP, which was considered to be quenched by Lys120 (*Staiano et al., 2007*). However, there are some differences in the conformation of Lys side chain in these proteins. In pOBP, the side chain of Lys120 is located practically parallel to the indole ring of Trp16, and the nearest atom of this residue NZ is separated from the center of the indole ring by only 4.15 Å. This conformation of Lys120 could be stabilized by H-bond formation between the O atom of main chain of Lys15 and NZ atom of Lys120 (2.63 Å, Fig. 7). In bOBP, the side chain of the Lys121 is extended and oriented in such a way that its NZ atom is located far from the center of the indole ring of Trp17. Therefore, it is unlikely that the Trp17 of bOBP is quenched by the Lys121. At the same time, Trp17 of bOBP can be quenched by electron transfer to the amide groups of the main chain. This mechanism works if there are negatively charged groups in the near vicinity of the indole ring and there are positively charged groups near the amides (*Callis, 2014*; *Scott & Callis, 2013*). In the case of Trp17 of bOBP, such situation is provided by Arg18 with its NH1 atom being H-bonded to the amide of the tryptophan residue (2.95 Å) and by Leu13 with its main chain O atom being H-bonded to the indole ring HN (2.87 Å). The Trp64 of bOBP makes a significant contribution to the total fluorescence of the recombinant bOBP. Indeed, in the microenvironment of Trp64 there are no groups which could promote electron transfer to amide groups of the main chain (Table 4). Additionally, the microenvironment of this Trp residue contains only a lysine residue (Lys59) in the conformation similar to that of Lys121, which does not favor fluorescence quenching.

The far-UV CD spectra recorded for the recombinant bOBP and its four mutant forms (bOBP-Gly121+, GCC-bOBP, GCC-bOBP-W17F, and GCC-bOBP-W133F) are rather similar and have a shape characteristic of protein enriched in the $\beta$-structural elements (Fig. 8). Decomposition of the far-UV CD spectrum of the recombinant bOBP using the

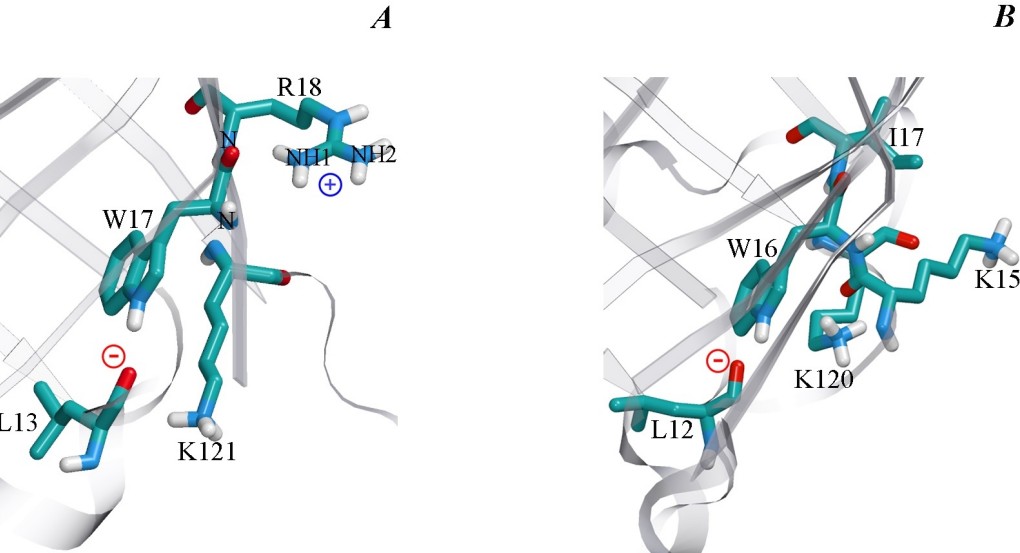

*A*                                                                                          *B*

**Figure 7  The microenvironment of Trp 17 in bOBP (A) and Trp 16 in pOBP (B).** The spatial orientation of lysine residues relative the indole ring of tryptophan residues is shown. The protein core is shown transparent. The drawing was generated based on the 1OBP file (*Tegoni et al., 1996*) and 1A3Y (*Spinelli et al., 1998*) from PDB (*Dutta et al., 2009*) using the graphic software VMD (*Hsin et al., 2008*) and Raster3D (*Merritt & Bacon, 1977*).

**Table 6  The evaluation of secondary structure of the recombinant bOBPwt and its mutant forms using Provencher's algorithm (*Provencher & Glockner, 1981*).**

|                    | $\alpha$ | $\beta$ | Turn  | Unordered |
|--------------------|-------|-------|-------|-----------|
| bOBPwt             | 0.133 | 0.359 | 0.204 | 0.297     |
| bOBPwt/OCT         | 0.113 | 0.354 | 0.207 | 0.311     |
| bOBP-Gly121+       | 0.085 | 0.400 | 0.208 | 0.303     |
| bOBP-Gly121/OCT    | 0.112 | 0.407 | 0.200 | 0.272     |
| GCC-bOBP           | 0.134 | 0.353 | 0.208 | 0.293     |
| GCC-bOBP/OCT       | 0.145 | 0.344 | 0.217 | 0.288     |
| GCC-bOBP W17F      | 0.077 | 0.415 | 0.209 | 0.299     |
| GCC-bOBP W17F/OCT  | 0.087 | 0.429 | 0.206 | 0.276     |
| GCC-bOBP W133F     | 0.154 | 0.356 | 0.206 | 0.269     |
| GCC-bOBP W133F/OCT | 0.172 | 0.352 | 0.200 | 0.275     |

Provencher's algorithm (*Provencher & Glockner, 1981*) revealed that this protein contains 13% $\alpha$-helix, 36% $\beta$-sheet, and 20% $\beta$-turns (Table 6). These data agree well with the results of the X-ray analysis of the wild type bOBP (13% $\alpha$-helix and 46% $\beta$-sheets). In the bOBP-Gly121+, adding the Gly121+ insert leads to a certain decrease in the content of $\alpha$-helical elements (Table 6). Obviously, the insertion of an extra Gly121 residue to the loop segment preceding the single $\alpha$-helix of the protein reduces the length of this helical element. On the other hand, introduction of a disulfide bond to the structure of the triple mutant GCC-bOBP leads to a full recovery of the protein secondary structure. Since the introduced disulfide bridge tightens the $\alpha$-helical region to the $\beta$-barrel frame of the protein, this tightened overall fold may be the reason for

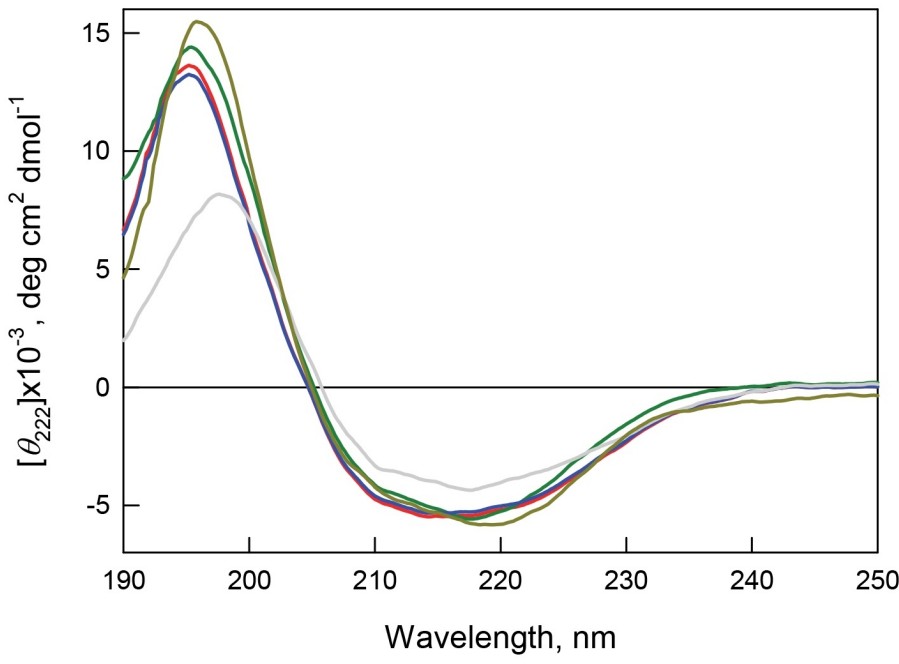

**Figure 8** Secondary structure for bOBP (red) and its mutant forms bOBP-Gly121+ (green), GCC-bOBP (blue), GCC-bOBP-W17F (gray) andGCC-bOBP-W133F (dark yellow) in buffered solution are indicated by far-UV CD spectra.

the restoration of the bOBP secondary structure. These data also confirm the stabilizing effects of the disulfide bond in the monomeric form of the protein. Replacement of the tryptophan residue Trp133 to phenylalanine in the mutant form GCC-bOBP-W133F leads to an increase in the content of $\alpha$-helical elements while the content of $\beta$-sheets stays unaltered. Instead, the amount of unordered structure is lowered. It is likely that the substitution of the tryptophan residue with a bulky side chain by a less massive residue in the single $\alpha$-helix of this protein may leads to prolongation of the helical element reducing some steric constraints inside the $\alpha$-helix. In contrast, in the GCC-bOBP-W17F mutant, replacement of a tryptophan residue Trp17 to phenylalanine may indicate a marked decrease in $\alpha$-helical structure and marked increase in $\beta$-structure with no changes in unordered structure content. We hypothesize that the absence of the bulky side chain of Trp17 in the first $\beta$-strand of the $\beta$-barrel may favor the formation of a closer contact between the first and the ninth $\beta$-strands of the protein thus stabilizing $\beta$-barrel.

In the presence of the OCT ligand, the fluorescent characteristics and the near- and far-UV CD spectra of the recombinant bOBP and its mutant forms undergo significant changes, indicating compaction and stabilization of the spatial structure of these proteins (Tables 1 and 6). These data also suggest that all mutant forms of bOBP analyzed in this work retain the ability to bind a natural ligand.

Therefore, all mutant forms of bOBP generally retain tertiary and secondary structure. Their structural organization is rather similar to that of the recombinant bOBP. The structure of the GCC-bOBP is closest to the structure of the recombinant wild type bOBP.

The observed changes in the local structure of the mutant forms of bOBP do not violate the ability of the protein to correctly fold and bind a natural ligand. Ligand binding to the bOBP mutant forms leads to a more compact state of the studied proteins and does not alter its oligomeric status.

## CONCLUSIONS

We show here that the insertion of Gly121+ leads to disruption of the domain swapping mechanism, resulting in a stable monomeric mutant protein bOBP-Gly121+. The introduction of a disulfide bond induces noticeable stabilization of the monomeric fold of the GCC-bOBP mutant. The amino acid substitutions introduced to bOBP in this study, such as Gly121+ insertion in the bOBP-Gly121+ mutant, replacement of the Trp64 and His156 to the cysteine residues in the GCC-bOBP mutant, and replacement of the Trp17 and Trp133 residues to phenylalanine in the GCC-bOBP-W17F and GCC-bOBP-W133F mutants, do not disrupt the functional activity of the protein. We show that the ligand binding leads to the formation of a more compact and stable state of the recombinant bOBP and its mutant monomeric forms. We also describe the peculiarities of the microenvironment of tryptophan residues of the protein which are essential for the formation of the fluorescent properties of the protein and which were not described previously.

**Abbreviations**

| | |
|---|---|
| **bOBP** | bovine odorant-binding protein |
| **pOBP** | porcine odorant-binding protein |
| **GdnHCl** | guanidine hydrochloride |
| **CD** | circular dichroism |
| **UV** | ultra-violet |
| **Parameter** $A$ | $(I_{320}/I_{365})$ upon excitation at $\lambda_{ex} = 297$ nm |

### Funding

This work was supported in part by the Program "Molecular and Cell Biology" of the Russian Academy of Sciences (KKT) and a grant from St. Petersburg in the field of scientific and technological activities (Olga S). The funders had no role in study design, data collection and analysis, decision to publish, or preparation of the manuscript.

### Grant Disclosures

The following grant information was disclosed by the authors:
Program "Molecular and Cell Biology" of the Russian Academy of Sciences (KKT).
Grant from St. Petersburg in the field of scientific and technological activities (Olga S).

### Competing Interests

Irina M. Kuznetsova, Vladimir N. Uversky and Konstantin K. Turoverov are Academic Editors for PeerJ.
## Author Contributions

- Olga V. Stepanenko performed the experiments, analyzed the data, wrote the paper, prepared figures and/or tables, reviewed drafts of the paper.
- Denis O. Roginskii and Olesya V. Stepanenko performed the experiments, analyzed the data, prepared figures and/or tables, reviewed drafts of the paper.
- Irina M. Kuznetsova and Konstantin K. Turoverov conceived and designed the experiments, analyzed the data, wrote the paper, reviewed drafts of the paper.
- Vladimir N. Uversky conceived and designed the experiments, analyzed the data, wrote the paper, prepared figures and/or tables, reviewed drafts of the paper.

## Data Availability

All generated data are adequately represented by figures and tables incorporated into the manuscript.

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
