# Peer review of "Structure and stability of recombinant bovine odorant-binding protein: I. Design and analysis of monomeric mutants"

_PeerJ, doi:10.7717/peerj.1933_

## Round 0.1 · original submission · Major Revisions

Both reviewers stress that some of your conclusions are too speculative or do not take into account alternative interpretations (e.g. regarding the influence of the micro-environment of the Trp residues, etc). Please address those concerns.

A few personal comments:

The references to PONDR(R) VLXT and PONDR VSL2 (lines 173-174 and 183-184) are wrong: neither Stepanenko et al. 2015 nor Stepanenko et al. 2014 contain any mention of PONDR(R).

legend to figure 1: "drown in green" should be either "drawn in green" or "shown in green"

line 227 "This indicates that the amino acid substitutions introduced to the bOBP sequence did not affect the compact structure of this protein." Were there reasons to expect such effects?

line 327 "a balky tryptophan residue " should be "a bulky tryptophan residue"


"Giordano A, Russo C, Raia CA, Kuznetsova IM, Stepanenko OV, Turoverov KK, Stepanenko OV, Kuznetsova IM, Turoverov KK, Huang C, and Wang CC. 2004. Highly UV- absorbing complex in selenomethionine-substituted alcohol dehydrogenase from
Sulfolobus solfataricus Conformational change of the dimeric DsbC molecule induced by GdnHCl. A study by intrinsic fluorescence. J Proteome Res 3:613-620."

should be

"Giordano A, Russo C, Raia CA, Kuznetsova IM, Stepanenko OV, Turoverov KK, Stepanenko OV, Kuznetsova IM, Turoverov KK, Huang C, and Wang CC. 2004. Highly UV- absorbing complex in selenomethionine-substituted alcohol dehydrogenase from
Sulfolobus solfataricus J Proteome Res 3:613-620."

Reviewer 1 ·

Basic reporting

.

Experimental design

.

Validity of the findings

.

Additional comments

Comments on manuscript entitled: “Structure and stability of recombinant bovine odorant-binding protein: I. Design and analysis of monomeric mutants”


• Lines 68 and 69: “Curiously, despite rather high sequence identity between porcine and bovine OBPs (42%), these lipocalins are characterized by different quaternary structures,”

These lines are to be changed. There is no relation between primary and quaternary structures.

• Lines 129 to 131: “Fluorescence lifetime were measured using a “home built” spectrofluorometer with nanosecond impulse (Turoverov et al. 1998) as well as micro-cells (101.016-QS 5 x 5 mm; Hellma, Germany)”.

I do not see any lifetimes and pre-exponential values. The authors indicate in Table 1 a mean fluorescence lifetime measured at one emission wavelength.

• Lines 238 and 239 : “recombinant bOBP is characterized by a relatively long-wave position of the intrinsic tryptophan fluorescence (λmax = 335 nm at λex = 297 nm).

No, 335 nm is a short wavelength position.

• Lines 247 to 254: “It should be noted that the side chains of the charged residues Lys121 and Lys59 included in the microenvironments of Trp17 and Trp64, respectively, are oriented parallel to the indole ring of the corresponding tryptophan residue, and their NZ amino groups are located at a short distance from NE1 group of the corresponding tryptophan residue (5.16 and 4.55 Å for NZ groups Lys121 and Lys59, Tables 3-4). Therefore, the presence of a partial fluorescence quenching of Trp17 and Trp64 cannot be excluded, since fluorescence quenching was previously reported for a single tryptophan residue Trp16 of porcine OBP that has similar features in its microenvironment (Staiano et al. 2007; Stepanenko et al. 2008).”

No, fluorescence intensity and lifetime modification are also observed for Trp residues in proteins even in the absence of Lys residues. Therefore, charge transfer is not necessarily correct.

• Are fluorescence spectra corrected for the inner-filter effect? Quantum yield is to be calculated for each protein, before drawing conclusions.


• The authors should update their fluorescence lifetimes and should write differently their paper by taking into consideration the work of Albani et al. on protein fluorescence and the newest interpretation given to fluorescence lifetimes origin.


The paper is to be rejected in its present form.

Reviewer 2 ·

Basic reporting

The submitted manuscript by Stephanenko OV et al “ Structure and stability of recombinant bovine odorant-binding protein: I. Design and analysis of monomeric mutants
conforms to PeerJ policies, contains a sufficient introduction and contains sufficiently referenced background literature.

The structure of the article conforms to PeerJ policies. However large parts of the introduction and methods hardly differ from the recent publication of the same authors (Stepanenko OV, Stepanenko OV, Staiano M, Kuznetsova IM, Turoverov KK, and D'Auria S. 2014b. The quaternary structure of the recombinant bovine odorant-binding protein is modulated by chemical denaturants. PLoS One 9:e85169.

Although this manuscript refers to several bOBP mutants, there is no detailed analysis on the mutation introduction on the plasmid’s sequences, primers used in methods and should be included. Instead the same information as the above reference is given.

The recommendation is that this work contains useful scientific information and should be published after modification in the results and discussion section in order to avoid repetition of results given by the same authors in recent publications and also to clearly correlate results data to atomic level conclusions. There are minor presentation changes.

Experimental design

The structure of the experimental design is valid and sufficiently described in some cases. However this manuscript refers to several bOBP mutants, with no detailed analysis on the mutation introduction on the plasmid’s sequences, primers used in methods and techniques used to introduce mutations. These should be included. Instead the same information as the above reference is given.

Some of the conclusions drawn are speculative since the method used (CD, fluorescence) are not able to distinguish information at atomic level.

Validity of the findings

The arguments analyzed in the Discussion section are closely related to the experimental findings although some suggestions are clearly not supported without more detailed evidence (e.g. structural studies).

In particular the support for argument at line 234 that OCT binding induces partial compaction because of the shift in the elution profile is not firmly based.
Also the changes in the far UV CD spectrum in the mutants (line 282) are attributed to changes in the regular secondary structure elements. In the case of Gly introduction this maybe obvious. In the case of the Cys-Cys introduction or the Trp replacement though the conclusions are highly speculative and not supported clearly from the experimental information.

There is no evidence from the experimental data how that the microenvironments of Trp17 and Trp133 (line 288) differ with each other and how this affects the fluorescence and near-UV CD spectra.

---

## Round 0.2 · Minor Revisions

The reviewer reports regarding your changes are very positive, but a few refinements should still be made. Please correct the legend of Figure 3, where faulty references for PONDR still remain. Please pay attention to the helpful considerations of (newly recruited) reviewer #3 regarding interpretation of fluorescence quenching.

Reviewer 2 ·

Basic reporting

This is a comeback from the first review.
Most comments suggestions of the referees have been corrected/explaned.
However remain some minor issues of overinterpretation of experimental data that can be resolved by rephrasing the arguments.

Experimental design

Satisfactory

Validity of the findings

354-364. There is no direct evidence and it is highly speculative to induce only from the decomposition of the far-UV CD spectrum on the increase/decrease of the helical/b-sheet content upon the Trp/Phe mutation.
357-358. “leads to” should be rephrased to “may indicate”, “It is likely that…leads..”
to “….may lead…”
363-364. “…favors…” Should be replaced by “may favour formation of closer contacts between the first ant the ninth b-strands of the protein introducing more stability to the b-barrel”.

Additional comments

Line 125 "analysed" instead of "analised"

Reviewer 3 ·

Basic reporting

No comment

Experimental design

No comment

Validity of the findings

See General comments for the author.

Additional comments

In general I find this manuscript acceptable, except for the language in lines 321-329

The suggestion that "Trp17 residue is substantially quenched by the Lys121" is actually quite tenuous, even though this may have gone unchallenged by previous reviewers. The authors do not seem to be aware of some very pertinent literature. Chen and Barkley and others have proven that the ammonium group will quench 3-methylindole (the best model for an absolutely unquenched Trp) by proton transfer in aqueous solution. But in a protein, this might or MIGHT NOT be happening even if the ammonium group ring somewhat close to the CZ2. In the crystal structure, the closest H on the ammonium is 5 Angstoms away, too far to transfer. Of course, proton transfer could happen because of the lysine is flexible and might come closer often enough to quench as it is in the crystal structure of porcine OBP. But there are a several proteins with closer NH3+ groups, e.g. the Trp in Staph nuclease which has two such lysines, that actually are closer to the ring on average, for which there is virtually no quenching. (the quantum yield is about 0.3). In fact there is little or no evidence that this kind of quenching happens in proteins at all, although it is well established in water. The point is you cannot be sure.

Probably more pertinent and more likely—but not mentioned—is that most intensity variations of Trp in proteins come from electron transfer from the indole ring to one of the two close amides. This has been shown to be switched on by having positive charge near the amide and/or negative charge near the indole. (See P.R. Callis / Journal of Molecular Structure 1077 (2014) 22–29 many references therein) Conversely such quenching is switched off by having negative charge near the amide and/or positive charge near the indole ring. Well, it should be noted in the manuscript that the Arg41, which has an H-bond to the Trp amide along with the O of Leu13 H-bonded to the indole ring HN (O is modeled with a -0.5 charge) is probably the more convincing reason that Trp17 is quenched . In summary, charged groups (with the exception of histidine cation) do not cause quenching; they create electric fields that can assist electron transfer quenching.

Thus, you now see that although the lysine might be quenching by proton transfer, it should be noted that there is good reason to consider electron transfer enabled by the position of charges and partial charges as well.

All of the above applies also to the Trp 16 of porcine OBP. There the Lys120, Arg40, and Leu12 all play the same roles. There the Lys NH3+ is in direct contact with the indole ring, but it probably doesn't stay that way all the time when in solution, as shown very nicely in Fig. 8 of Biochemistry 2007, 46, 11120-11127.

---

## Round 0.3 · accepted · Accept

Please note that lines 187-192 (which contained correct references for PONDR variants in v.1) have reverted to the wrong references present in v.0. The captions to figure 3 (which are correct in the manuscript body) are also not updated int the figure legends list. Please clear this up with the production team.